# Unraveling Axon Guidance during Axotomy and Regeneration

**DOI:** 10.3390/ijms22158344

**Published:** 2021-08-03

**Authors:** Miguel E. Domínguez-Romero, Paula G. Slater

**Affiliations:** Department of Cellular and Molecular Biology, Faculty of Biological Sciences, Alameda 340, Santiago 8331150, Chile; medominguez@uc.cl

**Keywords:** guidance cues, growth cone, axotomy, axon regeneration, spinal cord injury

## Abstract

During neuronal development and regeneration axons extend a cytoskeletal-rich structure known as the growth cone, which detects and integrates signals to reach its final destination. The guidance cues “signals” bind their receptors, activating signaling cascades that result in the regulation of the growth cone cytoskeleton, defining growth cone advance, pausing, turning, or collapse. Even though much is known about guidance cues and their isolated mechanisms during nervous system development, there is still a gap in the understanding of the crosstalk between them, and about what happens after nervous system injuries. After neuronal injuries in mammals, only axons in the peripheral nervous system are able to regenerate, while the ones from the central nervous system fail to do so. Therefore, untangling the guidance cues mechanisms, as well as their behavior and characterization after axotomy and regeneration, are of special interest for understanding and treating neuronal injuries. In this review, we present findings on growth cone guidance and canonical guidance cues mechanisms, followed by a description and comparison of growth cone pathfinding mechanisms after axotomy, in regenerative and non-regenerative animal models.

## 1. Introduction

The correct functioning of the nervous system is determined by the formation and maintenance of intricate networks of proper neuronal connections. During nervous system development, neurons extend an axon, which navigates through a complex embryo terrain, until finding its target. For this journey to occur, there must be good signage, and signals in the road must be well detected and integrated to reach the correct destination. Each growing axon presents at its tip a growth cone, which corresponds to a specialized highly motile, and cytoskeletal-rich structure, which senses the environment and guides the path for axon growth. Thanks to the differential expression of guidance cues receptors and the crosstalk between signaling pathways in the growth cone, the axon pathfinding is tunable. The environmental guidance cues bind their receptors present in the growth cone, activating signaling cascades that are integrated and result in the regulation of the growth cone cytoskeleton, defining growth cone advance, pause, turning, or collapse.

Fully elucidating the process of growth cone motility and regulation is of special interest for understanding and treating neurodevelopmental and neurodegenerative diseases, as well as neuronal injuries. After neuronal injuries in mammals, only axons in the peripheral nervous system (PNS) are able to regenerate to some extent, while axons from the central nervous system (CNS) fail to do so. The spinal cord is a very complex system, includes sensory and motor neurons, which need to extend very long axons to find their target, thus, it is an excellent model to study the role of guidance cues during neuronal development and regeneration.

Most studies have focused on determining the different extracellular cues, their receptors, signaling pathways, and how they influence the mechanisms of axon movement. These works have shown the existence of attractants and repellents molecules, which can be soluble or attached to cell membranes, defining long- and short-distance regulation, respectively. Although the attractive or repulsive effect of a specific molecule is not outright, during development, the axons encounter the co-existence of multiple types of guidance cues and they accurately respond to some of them; in addition, the same molecule can exert opposite effects in different axon types; furthermore, the same axon type can respond differently to the same guidance cue; and the ability of certain axons to respond to a specific molecule changes during development. Considering this complexity, the efforts have shifted towards the understanding of the context-specific regulation of guidance cues and their receptors, and several questions emerge. How are these interactions regulated? What happens after damage? Is there any differential regulation in regenerative (R) versus non-regenerative (NR) models? These questions need to be answered in order to improve therapies and induce axon regeneration. In this review, we will discuss some of the recent findings underlying the canonical guidance cues, Semaphorins, Netrins, Ephrins, and Slits, including their expression, signaling, and regulation during axon development and regeneration. We will start by giving an overview of the growth cone mechanisms allowing movement, followed by a description of the latest findings on growth cone guidance, and we will end with a description and comparison of growth cone pathfinding mechanisms after axotomy, in R and NR animal models.

## 2. Growth Cone: Leading the Way

Growth cones are specialized structures containing a sensory apparatus, a signal transducer device, and the motility machinery to guide and impulse axon growth. They are divided into three domains depending on the cytoskeletal distribution. The axon shaft widens to give rise to the central (C) domain, which is composed of mitochondria, vesicles, and microtubule (MT) bundles entering from the axon shaft and splaying out across the C domain. The further advance of these MTs is restricted of F-actin bundles forming F-actin arcs, which correspond to the transition (T) zone. In the most distal part of the growth cone, the peripheral (P) domain is found, which is composed by F-actin bundles forming filopodia, which are interspersed by actin mesh termed lamellipodia (Figure 1A). The F-actin bundles are usually polymerizing towards the leading edge of the growth cone, where they are the force pushing against the growth cone peripheral membrane, but at the same time, they present a retrograde movement towards the T zone, powered by myosin II, where F-actin is severed and dissembled (Figure 1B) [1,2]. Individual exploring MTs can be observed in the P domain, where they are in close interaction with F-actin, as they usually align to F-actin bundles, which serve as MTs polymerization guide (Figure 1C*i*), in addition, the MT movement is coupled to the F-actin retrograde flow, resulting in a drag of MTs towards the T zone (Figure 1C*ii*) [1,3]. This MT/F-actin coupling is fundamental for guided advance, as the interruption of this coupling by cytochalasin B induces the removal of F-actin bundles from the P domain and results in a randomized and uncontrolled entering of MTs to the P domain [3].

For the axon to advance, the growth cone follows three stages: protrusion, engorgement, and consolidation. During protrusion, the growth cone spreads a veil-like structure bounded by filopodia, towards the site of advance (Figure 1A) [4]. This structure mediates the formation of adhesive point contacts, consisting of the attachment of the growth cone cytoskeleton to the substrate through the generation of a protein complex “molecular clutch” (Figure 1D) [5]. In the engorgement stage, the molecular clutch is tightened up, and the generated tension results in decreased F-actin flow, and traction for growth cone translocation (Figure 1D*i*) [6]. At this point, the F-actin retrograde flow in the P domain is inverted and an anterograde flow towards the advance zone is observed, forming a corridor devoid of F-actin between the C domain and the substrate clutch. The F-actin arcs reorient and align with the direction of growth, delimitating the corridor through which organelles and MTs advance, filling the veil [7]. While the loss of F-actin retrograde flow also allows the protrusion of the growth cone beyond the clutch [7]. Finally, during consolidation, MTs are compressed into bundles to form a new segment of axon shaft [7].

Point contacts are formed when the extracellular matrix (ECM) ligands bind to their integrin receptors, which activates the assembly of a protein complex containing proteins such as talin, paxillin, and vinculin. This allows the activation of kinases like focal adhesion kinase (FAK) and Src [8], which are accompanied by activation of the Rho GTPase family (RhoA, Rac1, and Cdc42) signaling cascade, either directly by the adhesion of guidance cues to their receptors containing an intrinsic GTPase regulatory domain, or by modulating second messengers which act over GTPase regulatory proteins. The activated kinases lead to enriched levels of tyrosine-phosphorylated proteins, and recruitment of scaffolding proteins linking growth cone cytoskeleton with effector proteins that regulate cytoskeleton dynamics [8,9]. The pioneer MTs present in the P domain transiently uncouple from F-actin, allowing MTs polymerization towards the adhesion site [10], which play an instructive role by regulating kinases accumulation in the adhesion site [11], and together with the actomyosin-based tension-activated tyrosine kinase, drive F-actin accumulation and strengthening of the association F-actin/substrate [12]. Finally, once integrin/ligand binding is generated, more integrins are recruited for maximizing point contact maturation and traction force.

Point contacts are highly dynamic, and their formation, composition, size, and stability, depend on the substrate and Rho GTPases signaling, among others. Growth permissive substrates induce rapid axon extension and are characterized by the presence of highly motile growth cones, which generate many and dynamic point contacts [13] and are highly sensitive to changes in tyrosine phosphorylation signature [14]. In contrast, in less permissive growth substrates, axons extent at a lower velocity, form fewer contact points [13], and are less sensitive to changes in tyrosine phosphorylation [14]. In the case of Rho GTPase signaling, Rac1 activation is necessary for point contact formation, but in a window concentration-dependent way, as its inhibition blocks the formation, and its over-activation results in small and unstable point contacts [15]. Whereas activation of the RhoA target Rho kinase (ROCK), and inactivation of Rac1, are needed for stabilization and maturation of point contacts [15]. Additionally, myosin-II activity is regulated by ROCK and participates in the formation of point contacts, and therefore, its inhibition results in decreased number and size of them [15], and detachment of growth cone F-actin to coverslip substrate [2].

In addition to adhesive molecules and chemical signals, the mechanical forces to which the growth cone is subjected determine the growth cone movements as well [16,17]. For example, some axons, such as the ones from *Xenopus* retinal ganglion cells, are guided from stiffer to softer surfaces, showing straight and fast movement in stiffer surfaces, turning when exposed to stiffness gradients, and showing slower movement accompanied by increased growth cone area in softer surfaces [18], most probably due to changes in point contacts and thus growth cone traction forces. Not only do the number of recruited integrins define the maximal force, but the distance between integrins and the stiffness of the surface does as well. In softer surfaces, increasing the distance between receptors leads to increasing forces, while stiffener surfaces generate adhesion collapse [19,20]. The point contact dynamicity can be regulated downstream of attractive and repulsive signals, by FAK and Src activity. Point contact formation and turnover are regulated by the association between FAK and paxillin, which is affected by FAK phosphorylation [21]. Furthermore, growth cone advance correlates with contact point formation and decreased F-actin retrograde flow [5]. Fast-growing axons are characterized by the generation of many and dynamic point contacts [13] and reduced F-actin retrograde flow (Figure 1D*i*), compared to axons that extent at a slower velocity (Figure 1D*ii*) [5]. These modifications occur locally, as only F-actin bundles that are associated with point contacts show a reduction in F-actin retrograde flow [5]. Contrary to what occurs during growth cone extension, during growth cone retraction, F-actin retrograde flow is significantly increased [5].

During axon traveling, the growth cone advance is intercalated by repeated wandering periods, in which the growth cone pauses and senses the environment, defining the direction to follow and/or demarcating the location for collateral sproutings to be formed [22]. The growth cone pause is accompanied by morphological changes, in which the growth cone area increases, lamellipodia are largely spread, and MTs are contained in the C domain, showing a looped morphology, which must be re-organized into elongated bundles to resume advance [23]. This looped morphology is observed when the growth cone advance rate is decreased, and thus, the MT extension exceeds the axon growth rate. When the growth cone resumes advance, it can leave some remnants behind, which could be utilized for the formation of a future branch [24].

## 3. F-Actin and MT Cytoskeleton and Crosstalk

### 3.1. Actin Cytoskeleton

The actin cytoskeleton is a polarized structure, with the barbed end polymerizing at a greater rate than the pointed end. In the growth cone, the barbed end localizes towards the P domain, where polymerization in the leading edge pushes the membrane forward, resulting in protrusion; while the pointed end localizes towards the C domain, where F-actin severing happens (Figure 1B). F-actin dynamics is crucial for growth cone motility, therefore, actin-binding proteins regulating F-actin polymerization and severing are of special importance, e.g., F-actin located in filopodia, are important for environmental sensing; in point contacts, for traction generation; and in lamellipodia, for growth cone movement [25].

The most known F-actin polymerizing families are the actin-related protein 2/3 (Arp2/3) and formin (fmn), for nucleation of branched and unbranched filaments, respectively; and Ena/VASP (enabled/vasodilator-stimulated phosphoprotein) proteins, which bind to the F-actin barbed end, promoting polymerization of unbranched or sparsely branched F-actin. Therefore, Arp2/3 is of particular importance for lamellipodia structure, as it is formed by branched F-actin meshes, while fmn and Ena/VASP for filopodia, as they are composed by bundles of unbranched F-actin. Additionally, there is a group of proteins called capping proteins, which bind the F-actin barbed end, blocking the addition of new actin monomers, and thus, resulting in reduced length of F-actin. Finally, F-actin depolymerization and severing are mediated by the actin-depolymerizing factor (ADF)/cofilin family of proteins and the molecule interacting with CasL (MICAL) protein. ADF/cofilin mediates the dissociation of barbed end F-actin binding proteins, promoting F-actin depolymerization, which is followed by severing in small fragments, and recycling of G-actin, making it available for further polymerization. The action of ADF/cofilin generates new barbed ends, which can promote polymerization of the F-actin. In the case of MICAL, its depolymerizing activity is dependent on a redox enzymatic activity contained within the protein [25,26]. During growth cone protrusion and advance, actin nucleating, and barbed-end-binding proteins, as well as severing proteins, should be activated for generation of free barbed-end proteins, disposed of for protrusive polymerization [26].

### 3.2. MT Cytoskeleton

MTs are also polarized structures, with a rapid growing plus end localized towards the P domain. MTs stabilizing proteins, like MT-associated proteins (MAPs) and plus-end binding proteins (+TIPs), are needed towards the advancing or turning area, whereas MT destabilizing and severing proteins, such as stathmin and the kinesin 13 families, are also implicated in axon outgrowth and branching [26,27]. MAPs bind to MTs lattices for regulation of MTs dynamics. Examples of MAP proteins that participate in directed growth cone migration are Tau and MAP1B. Tau knockdown (KD) in hamster cortical neurons in culture induces disruption of MTs bundles, and, consequently, these MTs present disorganized trajectories and are unable to enter filopodia, resulting in shorter axons and failure of responding to repellent cues [28]. While MAP1B phosphorylation is needed for Netrin-induced growth cone response [29]. Additionally, the +TIPs bind to the MTs plus end of growing MTs, and they have been extensively related to directed growth cone response [30]. For instance, adenomatous polyposis coli (APC) accumulates in the MTs plus ends, in the side of growth cone turning [31], while APC2 induced MT stabilization is needed for responding to EphrinA2 [32]. The cytoplasmic linker-associated protein (CLASP) acts downstream of Slit/Robo signaling, through Abl phosphorylation regulation [33,34]. Additionally, the fly orthologue of *Xenopus* microtubule-associated protein 215 (XMAP215) presents a CLASP antagonistic role [33], even more, the XMAP215 binding partner transforming acidic coiled-coil protein 3 (TACC3), is also necessary for responding to Slit [35]. Moreover, both XMAP215 [36] and TACC3 [37] are needed for responding to the repulsive cue EphrinA5. Recently, it was also demonstrated the participation of the +TIP navigator-1 (NAV1) in growth cone turning induced by Netrin-1, in mouse cortical neurons [38].

In summary, both F-actin and MT dynamics are important for different aspects of axon elongation, F-actin is necessary for the mechanical aspect of the motility, while MTs define the direction of growth. Therefore, it is not surprising that the crosstalk between both systems is needed for the accurate response to guidance cues [9,39,40], which we are going to discuss in the following sub-section.

### 3.3. Actin/MT Crosstalk

The controlled and coordinated dynamics of F-actin and MTs determine growth cone shape and movement. F-actin and MTs coupling can be mediated by canonical F-actin or MT binding proteins, or protein complexes involving both F-actin and MTs binding proteins (Figure 1E).

The F-actin binding protein fmn has been described as one of the families of proteins that can exert the F-actin/MT association. Fmn2 localizes mainly throughout the F-actin bundles, but additionally, in some filopodia tips, and is needed for filopodia formation and elongation [41], as well as point contact stability and maturation [42]. This filopodia localization is required for guiding dynamic MTs in the P domain, stabilizing filopodia protrusion and growth cone directional movement [43]. The fmn disheveled-associated activator of morphogenesis (DAAM) can bind to MTs, both directly and indirectly, by binding the +TIP end-binding protein 1 (EB1), which is consistent with the often accumulation of DAAM in the MTs plus end. Additionally, DAAM has the ability to bind F-actin and MTs simultaneously in vitro, which together to the co-localization of F-actin/DAAM/MT in Drosophila primary neuronal cells, and the reduced MTs retrograde translocation in DAAM KO neurons, suggests a role of DAAM in F-actin/MT crosslinking, either by directly binding or through interaction with EB1 [44].

MT binding proteins are also candidates for mediating F-actin/MT coupling. One of the proposed families is the +TIP [45]. These proteins localize close to the guidance cue receptors and are subject to post-translational regulation downstream of guidance cues, influencing F-actin/MT association, and thus, many aspects of growth cone motility [30,45]. The +TIP cytoplasmic linker protein 170 (CLIP-170) associate with the fmn mDia1, and together are recruited to the MT plus end by EB1, resulting in acceleration of F-actin polymerization and increased duration of elongation in neuronal processes in rat primary cortical neurons [46]. This CLIP-170 effect over the F-actin elongation rate is not limited to mDia1, as it has been also observed for mDia2, DAAM1, and the N-terminally truncated Drf-like inverted formins (INF) 1 and 2 [46]. XMAP215 binds directly to F-actin in vitro and mediates F-actin/MT cross-linking in *Xenopus* spinal cord explants [36]. The latter is demonstrated by the co-localization of F-actin/XMPA215/MT in the growth cone periphery filopodia [36], the loss of MT distribution in co-linear tracks, and decoupling of MT from F-actin retrograde flow when knocking down XMAP215 [47], as well as decreased F-actin/MT alignment, and randomization of the directionality of exploring MTs, resulting in wandering growth cones not able to respond to directional movement [36]. NAV1 can also crosslink F-actin/MT in the growth cone periphery of cortical neurons. In vitro analysis showed that incubating NAV1 with F-actin and MTs induces F-actin/MT coalignment, and in mouse cortical neuron cultures NAV1 persists in the plus end of non-growing MTs, preventing the depolymerization of MTs in F-actin-reach regions. NAV1 loss resulted in reduced MTs in the growth cone periphery, increased MTs catastrophe events, and compromise to Netrin-1 growth cone steering [38].

As evidenced, F-actin/MT crosstalk is fundamental for growth cone-directed movement. Additionally, in the last few years, many proteins have emerged as F-actin/MT crosstalk regulators. Unraveling how these proteins are regulated by guidance cues, and their involvement during axon development and regeneration would be of great importance.

## 4. Canonical Guidance Cues: What Do We Know?

Semaphorins (Semas) comprise a large family of 30 proteins divided into eight classes, including secreted (class 2, 3, and V), transmembrane (class 1, 4, 5, and 6), and surface-attached proteins (class 7). Semas are usually considered as repulsive cues, although they have also been described to induce attraction [48]. This apparent contradictory function is determined by the substrate/receptor complex formed. Sema receptors are usually heterocomplex, including co-receptors. Additionally, a particular Sema can bind multiple receptors, resulting in different signaling cascades [49,50]. The canonical and most studied Sema receptors are Plexins, although they can heterodimerize with other receptors such as integrin receptors, and usually require the co-receptor Neuropilin for activating the corresponding signaling cascades [50].

Netrins are a family of extracellular laminin-related proteins, either secreted (Class 1, 3, and 4) or membrane-tethered (Class G1 and G2), which can bind two different receptors including deleted in colorectal cancer (DCC), its mammal paralogue Neogenin, and uncoordinated-5 homologs (UNC5A-D). The combination of receptors on the growth cone determines if the result corresponds to an attractive or repulsive one [51,52]. When Netrin binds to a DCC homodimer, or to a DCC/Neogenin heterodimer [53], an attractive response is observed, while, when it binds to a heterodimer composed by DCC and the co-receptor down syndrome cell adhesion molecule (DSCAM) results in axon branching [54]. Repulsion is observed when Netrin binds to a DCC/UNC5 heterodimer [55], while short-range repulsion is mediated when it binds to a heterodimer composed by UNC5 and the co-receptor DSCAM [56]. This attractive versus repulsive response can be in part explained by the ability of both DCC and UNC5 Netrin receptors to bind the highly dynamic β-tubulin isoform TUBB3, and thus, regulating MT dynamics. On one hand, exposure of mouse cortical neurons in culture to Netrin-1 induces DCC binding to TUBB3, capturing dynamic MTs at the Netrin-1 binding site and, therefore, promoting both MT dynamics and local stabilization, as well as the protrusion of the growth cone towards the Netrin-1 source [54,57]. On the other hand, UNC5/TUBB3 interaction is abolished by the exposure of mouse cerebellar neuron culture to Netrin-1, resulting in detachment of dynamic MTs and, therefore, MT growth increases further while decreasing near the source, where lamellipodia and filopodia are retracted [58]. Additionally, Netrin can generate a concentration-dependent bimodal response. Mice cortical neurons cultured in a passive microfluidic gradient chamber showed a Netrin attractive response when exposed to a low concentration gradient, and a repulsive response when exposed to a high concentration gradient [59]. Interestingly, the exposure to these different concentration gradients does not affect the expression or localization of DCC nor UNC5 [59], suggesting that Netrin concentration could be a novel form of response-regulation, independent of the receptor expressed. Furthermore, Netrin has been described to exert its function locally and in gradients. Even through shallow gradients, Netrin is able to induce highly polarized shootin1 phosphorylation towards the origin source, promoting its binding to the L1 cell adhesion molecule (L1-CAM) [60]. Additionally, shootin1 can bind directly to the F-actin binding molecule cortactin, and this association is enhanced by shootin1 phosphorylation [61]. These interactions increase traction force and decrease F-actin retrograde flow, allowing Netrin-1–induced axon guidance [60,61].

Slits are a family of three secreted glycoproteins in vertebrates, Slit1, Slit2, and Slit3, which are subjected to a proteolytic process, generating an N-terminal fragment (SlitN) and a C-terminal fragment (SlitC), that bind to Robo and Plexin A receptors, respectively [62]. Additionally, FLRT3 is a Robo1 co-receptor, which is needed for modulating the Netrin-1 attractive response [63].

Ephrins comprise a large family of membrane-tethered ligands for Eph receptors, with numerous functions during development—axonal guidance, and during adult life—wound healing. Eph is the largest family of receptor tyrosine-kinases (RTKs), with 14 receptors divided into two subfamilies, EphA, which binds EphrinAs, and EphB, which binds EphrinBs. Although, some irregularities have been observed, as EphA4 that is able of binding class B Ephrins, or EphB2 that also binds EphrinA5. The signaling through these cues and their receptors is based on cell-cell contact, and the ligand-receptor interaction triggers bidirectional signaling, both from the ligand and from the receptor [64].

Although the canonical guidance cues are well characterized, their function and regulation are rather complex. These molecules exert their effects in a context-specific dependent way and can influence each other, explaining how multiple and simultaneous guidance cues inputs are integrated, resulting in the correct axonal growth steering. In the following sections, we will discuss some examples.

## 5. Guidance Cues in Concert

It has been observed that crosstalk between the families of canonical guidance cues is necessary for defining sensibility to each other. For example, in the thalamocortical area (TCA) system, the rostral TCA is sensitive to Netrin-1, while intermediate TCA is not [63]. This sensibility to Netrin requires the up-regulation of DCC in the plasma membrane, which is achieved by a simultaneous exposure to Slit and Netrin, and which is also dependent on the Robo1 co-receptor FLRT3. Suggesting a change in the Slit/Robo1 signaling cascade, which only occurs in the rostral TCA nestin-responsiveness axons [63]. Additionally, promiscuity of the guidance cues family members exists, as it is known that some receptors bind different ligands, including ones with opposite effects. For example, when Neogenin binds Netrin-1, it elicits an attractive response, but when Neogenin binds the repulsive guidance molecule (RGM), it mediates a repulsive response. Interestingly, these two ligands, instead of competing for Neogenin binding, bind to the receptor at the same time, forming a ternary complex, which acts as a silencing complex, inhibiting both the attractive and the repulsive response [65]. In the present section, we will give an overview of the complexity and importance of guidance cues in families crosstalk, by the comparison of guidance cues behavior in embryos versus adults, as well as the role of guidance cues during the spinal cord development and their role on cytoskeleton regulation.

### 5.1. Guidance Cues in Embryonic Versus Adult Nervous System

The complex regulation of sensibility to different guidance cues is exemplified by the comparison between embryonic and adult axons. Both embryonic and adult axons are susceptible to Semas, although they do not respond to them in the same way [48,66]. They have differential expression patterns that influence the response, for instance, the atypical intermediate filament nestin, which is expressed in neuronal progenitors and newborn neurons, localizes mainly to the distal area of growing axons in embryonic, but not in adult neurons [67]. Neurons expressing nestin are more sensitive to Sema3A than the ones that do not express it [67], and this effect is dependent on Doublecortin (DCX) [68]. Nestin binds DCX and the cdk5 kinase activator p53, facilitating DCX phosphorylation by cdk5, a kinase that acts downstream of Sema3A [68]. Interestingly, studies in *Xenopus laevis*, comparing the transcriptomic response after spinal cord injury (SCI) in R versus NR stages, showed a differential increase in nestin transcripts in NR-stages, while no changes in R-stages, at 6 days post-injury (dpi) [69,70]. The timing of this response coincides with axon bundles crossing the injury gap in R-stages, and the lack of them, but the formation of a glial scar in NR-stages [70,71]. Additionally, although Sema3A has been described mainly as a repulsive guidance cue, recent studies have suggested an opposite Sema3A effect in the embryonic versus the adult nervous system. While axons from embryonic dorsal root ganglia (DRG) neurons in culture show axonal growth cone collapse and axon retraction in response to Sema3A, axon growth is induced in adult DRG and trigeminal ganglia (TG) neurons in culture, as well as cornea nerve regeneration in vivo, after Sema3A treatments [66]. Accordingly, this observation has also been reproduced in the CNS, specifically in the cortical tissue in a traumatic brain injury (TBI) model. Implantation of a hydrogel containing a Sema3A gradient, in the site of rat cortical brain injury, induced cortical regeneration [72]. The same result was observed with a hydrogel containing a Netrin-1 gradient [72]. This evidence, added to the increase of Sema3A in the cells within the glial scar, leads to the question of whether the increase in nestin and Sema3A work together to sensitize the axons to Sema3A, generating axonal repulsion, and thus impeding axon regeneration in mammals and NR animals.

### 5.2. Guidance Cues and Spinal Cord Development

In some cases, axons need to navigate over long distances, for instance along the spinal cord [73]. Usually, in these cases, an intermediate target guides the way, by attracting the axons until contacting the intermediate target, which is followed by a change in repulsion, so axons can reach their destination. This fine-tuning needs a switch in responsiveness, which can be achieved by changes in the guidance cues present in the intermediate target, changes in the receptors expressed in the axon growth cone, or by crosstalk between guidance cues. For instance, in the spinal cord, dorsal horn resident axons navigate until they reach the intermediate target, the floor plate, where the axons cross the midline and turn rostrally. During the navigation towards the floor plate, the Sema3B receptors Neuropilin-2 and Plexin A1 are expressed, although Plexin A1 is processed by calpain, resulting in no responsiveness to Sema3B. In the floor plate, calpain activity is suppressed, and the axons start to respond to Sema3B [74]. Similarly, corpus callosum axons grow towards the brain midline, and after crossing it, they follow the contralateral hemisphere. During navigation, axons are attracted towards the brain midline by Sema3C, and they lose responsiveness to Sema3C after midline crossing because of upregulation of Ephrin-B1, which can form a protein complex with the Sema3C receptor Neuropilin-1, blocking the Sema3C signaling [75].

During the formation of the corticospinal circuits, the axons extend along the spinal cord and send collateral projections to the target motor neurons. Motor neurons exhibit a common ventral trajectory towards the floor plate until they reach the limb level, where the trajectories are separated. The medial motor column (MMC) neurons navigate dorsally to innervate muscles in the back, while lateral motor column (LMC) neurons continue towards the limb, where they divide into the dorsal or ventral compartment to innervate extensor and abductor, or flexor and adductor muscles, respectively [76]. The ventral trajectory of axons is guided by Netrin-1 derived from ventricular zone located neuronal precursors, which accumulates in the two-thirds of the ventral portion, on the pial surface [77,78]. Netrin-1 is needed for axons confinement in the spinal cord during the ventral migration, avoiding premature emigration [79]. The ventral spinal cord expresses Sema3A, while the corticospinal axons express Neuropilin-1, therefore, the collateral projections avoid entering the ventral spinal cord due to Sema3A-Neuropilin1 signaling [80]. Additionally, during the development of the corticomotoneuronal (CM) circuit, transient ipsilateral CM connections are observed, which are eliminated in a Sema6D-PlexinA1 signaling-dependent way [81,82]. The non-apoptotic Bax/Bak-caspase-9 pathway, is necessary for this circuit refining process [83], a pathway that is neuronal activity-dependent and regulates PlexinA1 expression [82].

Axons involved in limb-innervation of vertebrate spinal LMC neurons are responsive to Netrin-1, but only when they are also exposed to the repellent cue Ephrin-A5. Although this effect is observed in only a subpopulation of axons, Ephrin-A5 does not exert the same effect on the ones from dorsal spinal cord neurons [84]. In chick spinal cord explants, Ephrin-A5-EphA4 binding results in increased levels of the Netrin receptor Neogenin in the plasma membrane, as well as Neogenin/EphA4 co-localization, and Netrin-1 binding to the growth cones [84]. For axons to leave the spinal cord and innervate the limb muscles, they need to suppress responsiveness to Netrin-1, otherwise, they are attracted towards the pial surface. This suppression depends on the Rho GTPase regulator p190RhoGAP, which binds the Netrin receptor DCC, changing its location towards the growth cone P domain [85].

All the above-mentioned evidence demonstrates the complexity and tangling of the guidance cues signaling. Therefore, it is necessary to study the different guidance cues in a context-dependent way.

### 5.3. Guidance Cues and Cytoskeleton Regulation

Guidance cues exert their attractive or repulsive function by modulating cytoskeleton dynamics. For example, Semas exert their repellent function by generating growth cone collapse, which is achieved by modulation of F-actin and MT dynamics, and contact point stability. When a growth cone is exposed to Semas, loss of F-actin in the periphery and reduced F-actin polymerization capacity are observed, which are mediated by the multidomain oxidoreductase enzyme MICAL [86]. These effects over F-actin can be attained due to direct binding of the cytoplasmic portion of the Sema receptor Plexin to MICAL and, additionally, Plexin is also able to bind different members of the GPTases family, leading to the regulation of F-actin binding proteins [86]. For achieving MT dynamics regulation, Sema/Plexin signaling leads to the activation of multiple kinases, which phosphorylate and activate collapsin response mediator proteins (CRMP), which bind tubulin and regulate MT dynamics [87]. Finally, Sema/Plexin signaling regulates integrin function, by decreasing its expression or activation, resulting in the disassembly of cell-substrate adhesion [88].

Studies focusing on how guidance cues regulate the growth cone cytoskeleton for generating an attractive or repulsive response have opened more questions about the complexity of guidance cues regulation. Different repulsive guidance cues can regulate cytoskeletal dynamics differently, and guidance cues with opposite effects can generate a similar initial cytoskeleton regulation. For example, Slit gradients induce growth cone lamellipodia retraction and filopodia elongation towards higher concentrations of Slit. This response seems to be contrary to what would be expected for a repulsive cue, but surprisingly it is necessary for a repulsion net response away from Slit both in vitro and in vivo [89]. This effect is dependent on the formation of a complex between the Slit receptor Robo-1 and the Ena-VASP protein Mena, an F-actin binding protein that promotes actin filament elongation [89]. Filopodia elongation is not necessary for Sema3A induced repulsion in vitro [89]. Whereas for Netrin induced attraction the mechanism involves filopodia elongation, which implicates the participation of the protein VASP and its regulation by monoubiquitination [90].

## 6. Guidance Cues in Axotomy and Axon Regeneration

Most of what we know about the role of guidance cues in the steering of axonal growth is based on studies during neuronal development. Although, what happens with guidance cues, or how they contribute after axonal damage, has not been widely studied. Therefore, the elucidation of the role of guidance cues underlying a successful or failed axon regeneration, is of special interest, as it could provide new targets for therapeutic treatments for improving axon regeneration. In this review, we provide an overview of guidance cues in axon damage and regeneration, with a focus on SCI.

### 6.1. Spinal Cord Injury

SCI affects nerves involved in receiving sensory information and controlling motor response, and thus, it results in paralysis beneath the injury site. Additionally, after SCI, some internal organs are disconnected from central nervous system regulation. Due to the limited regenerative capacity of the spinal cord in humans and mammals in general, and the absence of therapies allowing functional and efficient recovery, this affliction is permanent. SCI consists of a rapid mechanical insult, usually unexpected and almost impossible to intervene, causing direct disruption of axons, and oligodendrocyte death. The biochemical changes generated in this primary injury are propagated, leading to a secondary injury phase characterized by increased cell death and dysfunction, which can last for months to years [91]. During this period, a glial scar is formed, which protects from further neuronal loss and tissue degeneration [92,93]. After axotomy in mammalian CNS, the axons are not able to regenerate; interestingly, there are some R models, such as mammalian PNS and non-mammalian animals. PNS regeneration is generally slow and results in poor functional recovery. In contrast, non-mammalian animals such as teleost fishes, urodele amphibians, and larval stages of anuran amphibians, have extraordinary regenerative capacities, showing functional recovery. In the following sub-sections we will give an overview of three important events that occur differently in R versus NR SCI models: (1) growth cone formation; (2) inflammatory response, and (3) glial scar or glial bridge generation (Figure 2).

#### 6.1.1. Growth Cone Formation

After axotomy in the mammalian CNS, the proximal axon extends its main process or generates collateral sprouting, even after the glial scar formation [94,95]. Nevertheless, the axons seem to lack directionality [96], and end up in proximal axon end swelling, forming a terminal bulb, which is a very dynamic structure, with an unusual arrangement of its cytoskeletal network [97] that guides axon degeneration by constantly retracting away from the injured site [98] (Figure 2A). Interestingly, axon degeneration is dependent on MT destabilization, as low doses of the MT stabilizing drug Taxol prevents the formation of retraction bulb [99] and reduces glial scarring [100] while increasing axon re-growth. In contrast, in the PNS [99] and in regenerative animal models [101], motile growth cones are formed at the tip of the axons, which guide the regeneration (Figure 2A,B).

After damage, the membrane of the proximal axon end must be sealed, the cytoskeleton re-assembled, and transport of molecules and local translation need to occur to generate a growth cone and allow axon regeneration [102]. All these processes are highly energetically demanding and, thus, require the presence of functional mitochondria in the growth cone. Augmented mitochondrial motility towards the injury site results in an increase in the number of axons forming a growth cone and in improved axonal regenerative capacities [103,104].

Mitochondrial fission/fusion balance and function seem relevant for responding to guidance cues and growth cone steering [105]. Many molecules that act as attractive and repulsive molecules influence mitochondrial function regulation, including chondroitin sulfate proteoglycans [106], NGF [107], and BDNF [108]. In addition, the canonical repellent guidance cue Sema3A generates an increase in mitochondrial membrane potential [107]. After axotomy in mammals, mitochondrial dysfunction, and decreased mitochondrial motility are observed, leading to a huge deficit of energy [109,110], which is in accordance with failure in growth cone generation, and the presence of a terminal bulb and axon degeneration, as well as the lack of directionality of axon sprouts [96]. On the contrary, in regenerating axons, an increase in mitochondrial motility is observed, which is required for the replacement of damaged mitochondria [111] and correlates with growth cone formation and axon regeneration. All these studies suggest an important role of mitochondria in growth cone directional steering and would be interesting to define if functional mitochondria could compensate for the lack of directionality and regeneration observed in the mammalian CNS [96].

#### 6.1.2. Immune Response: Pro- and Anti-Inflammatory Environment

Another difference observed between R versus NR models is the inflammatory response. Usually, a pro-inflammatory response is activated for cell debris removal, but at the same time, the pro-inflammatory cytokines induce cellular damage, increasing immune cell infiltration and extracellular matrix damage. This early response must be followed by an anti-inflammatory response, for resolving inflammation and for tissue remodeling and repair to occur [112]. In the mammalian CNS, PNS, and in regenerative animals, a pro-inflammatory (neurotoxic) response is activated after injury. In mammalian CNS, this pro-inflammatory response is followed by a transient anti-inflammatory (neuroprotective) response, that is finally overtaken by a pro-inflammatory response, leading to chronic inflammation [113]. In the mammalian PNS, the pro-inflammatory response is followed by a mixed pro-anti-inflammatory phenotype or anti-inflammatory response [114]. Finally, in regenerative animal models, such as zebrafish, the pro-inflammatory response is followed by a sustained anti-inflammatory response [115]. The inflammatory response also correlates with the capacity of cell debris clearance. Mammalian CNS microglia present a limited phagocytic capacity, and after injury, an amplified microglia response is observed, while both mammalian PNS and regenerative animals present an efficient cell debris clearance [101].

Several studies have evaluated the role of neuronal guidance cues on immune response modulation, such as immune cells interaction, migration, infiltration, and cytokines production or secretion [116,117]. Herein, we focus on the evidence showing the involvement of different guidance cues in defining the immune cells’ polarization towards pro- or anti-inflammatory phenotype.

The Sema family shows mostly a pro-inflammatory role, with exception of Sema4D, which presents s pro- or anti-inflammatory role depending on the cellular context. In cultures of primary mouse microglia, Sema4D causes an increase of nitric oxide (NO) synthase expression and NO production [118], which is characteristic of activated microglia, therefore suggesting a pro-inflammatory role. In contrast, in lipopolysaccharide (LPS)-activated microglia, which presents an anti-inflammatory phenotype, Sema4D induces a resting phenotype, with lesser migratory capability and NO production [119]. Sema4A is highly expressed in dendritic cells in experimental autoimmune encephalomyelitis (EAE), an autoimmune disease in which immune cells degrade myelin nerves [120], and increase Th17-mediated neuroinflammation [121]. Sema4B seems to act as a pro-inflammatory signal in vitro and after brain cortex injury, as it is fundamental for proper astrocyte activation and proliferation [122]. Finally, Sema7A increases the inflammatory cells infiltration after corneal nerve injury, which is evidence of a pro-inflammatory role [123].

Conversely, most evidence supports an anti-inflammatory function for Netrin-1 after nervous system injury. Netrin-1 signaling through UNC5B receptor inhibits microglia activation and decreases pro-inflammatory cytokine levels after adult rat brain injury [124]. Additionally, Netrin-1 KD leads to an in vivo increase in activated astrocytes after middle cerebral artery occlusion in mice [125]. A reduced expression of Netrin-1 is observed in the spinal cord and cerebellum of EAE mice, which presents a pro-inflammatory environment, further supporting the Netrin-1 anti-inflammatory role [126]. Furthermore, a pro-inflammatory stimulus such as tumor necrosis factor (TNF) and interferon-gamma (IFNγ) on primary cultures of human blood-brain barrier (BBB) endothelial cells, causes an upregulation of Netrin-1, which then exerts an anti-inflammatory function by reducing pro-inflammatory cytokine secretion [127].

Ephrin has been mostly associated with pro-inflammatory response, EphA2 is involved in pathogen recognition and pro-inflammatory response activation in non-neuronal [128] and in BBB epithelial cells [129]. A transcriptomic profile comparison showed that treatment of astrocytes with EphB1 generates an up-regulation of pro-inflammatory genes [130]. In the case of EphA, evidence shows a pro- and anti-inflammatory role, thus, its effect on immune cells polarization is not entirely decided [131,132,133].

Finally, an anti-inflammatory role for Slit2 has been suggested. On adult rat injured brain, Slit-2 reduces pro-inflammatory cytokines expression [134] and increases BBB tight junction proteins expression, improving its impermeability to infiltrating immune cells [135]. Additionally, overexpression of Slit-2 inhibits astrocyte activation in the brain parenchyma and maintains the integrity of the BBB in the aging brain [136].

Considering that a pro-inflammatory environment is defined as neurotoxic, and an anti-inflammatory as neuroprotective, and the differences in the inflammatory response between R and NR animals, it would be interesting to determine if modifying the guidance cue signature after damage changes the regenerative capacity.

#### 6.1.3. Glial Scar

In mammalian CNS, the chronic inflammation leads to a glial scar generation [137], while in mammalian PNS [138] and in R models [139] this scar is not observed, but instead, a glial bridge is formed in the injury area, guiding the trajectory for axon regeneration.

The glial scar formed in the lesion site is composed of microglia, astrocytes, inflammatory cells, meningeal fibroblasts, pericytes, and extracellular matrix [137,140]. The most studied cells are astrocytes, which aggregate in the borders of the injury, and fibroblasts, which locate in the center of the scar and express mainly repulsive guidance cues. Additionally, some cells express molecules that weaken the growth cone point contacts, resulting in inhibition of neurite elongation [141]. For several years it has been considered that the glial scar function is to contain and stop damage dissemination, but at the same time acts as a physical and chemical impediment for axonal regeneration, due to the inhibitory environment that creates. Nevertheless, recent studies have challenged this view by showing that growth factor supplemented axons are able to grow through the glial scar and that some glial scar cells produce and secrete molecules that promote axonal regeneration [142,143].

It has been demonstrated that guidance cues, mainly Semas and Ephrins, are associated with the glial scar (Figure 2A). Sema3A, and its receptor, Neuropilin-1, were the first to be proposed as contributors to glial scar formation [144,145]. Sema3 accumulates in the fibroblasts present in the center of the glial scar, contributing to the repulsive environment that impedes the migration of Neuropilin-1 positive regenerating axons through the injury site [145,146,147]. This repulsive role is supported by studies of Sema3A inhibition, which show improved regeneration and functional recovery after SCI [148], and suppress activation of astrocytes in vitro, after glucose-oxygen deprivation, and in vivo, after middle cerebral artery occlusion [149]. Other Semas that have been associated with the glial scar formation are Sema7A, which is highly expressed in reactive astrocytes accumulated in the injury site of adult rat spinal cord [150], and the transmembrane Sema4B, which is expressed in reactive astrocytes of mice with an injured cerebral cortex, and is required for astrocyte activation and proliferation [122]. This evidence suggests the role of Semas in glial scar formation, promoting astrogliosis by favoring astrocytes proliferation and activation, and providing the repulsive environment for axonal navigation.

The association of Ephrins and Eph with the CNS glial scar relies mainly on the EphrinB2/EphB2 axis. In adult rat SCI, EphrinB2 is expressed in astrocytes, and EphB2 in meningeal fibroblast, within the glial scar [151,152]. Transgenic mice with EphrinB2 conditional deletion restricted to astrocytes present lesser astrogliosis, reduced glial scar, an increased number of corticospinal axons penetrating the injury area, and a better motor function recovery after SCI [153]. This is also supported by both an in vitro study, in an astroglial-fibrotic scar model induced by a co-culture of astrocytes and fibroblasts treated with transforming growth factor ß1 (TGF-ß1) [154], and an in vivo study after adult rat SCI [152], in which Ephrin-B2 RNAi silencing results in reduced astrocyte-fibroblast aggregation and improved axonal traveling through the remaining astroglial-fibrotic scar in the culture, or through the injury area after SCI. EphA4 has also been associated with glial scar formation both in vitro and in vivo. In vitro studies show that neurons cultured on an EphA4−/− astrocyte monolayer extend longer neurites compared to WT [155], and that EphA4 inhibition during plaque scratch injury, reduces reactive astrocytes accumulation within the scratch [132], while EphA4 activation with EphrinA5-Fc enhances astrocytes migration and proliferation [156]. An in vivo analysis showed co-localization of EphA4 with glial fibrillary acidic protein (GFAP)+ astrocytes in the lesion site after adult mice SCI [131,155], and in adult marmoset monkey brain injury [156].

Additionally, PlexinB2 is upregulated in activated microglia/macrophages after adult rat SCI and is necessary for corralling, a process involving immune and glial cells movement to the injury core for sealing the wound [157]. Finally, Sema4D is transiently upregulated after SCI, principally in oligodendrocytes [158,159]. Considering that Sema4D acts as a repulsive cue for mature axons [158] and that KD:Sema4D animals present better locomotor recovery during 8 weeks after SCI [159], it is tempting to associate this class 4 semaphorin with an inhibitory role during CNS regeneration.

Finally, even though there are scarce studies on Netrins and Slits, expression of these cues is detected in the center of CNS lesions, in the cerebellum, spinal cord [160], and brain [161]. Netrin-1 expression within the glial scar is associated with activated microglia and macrophages, while the UNC5H receptor is associated with fibroblast and macrophages [160]. Slits expression is also associated with GFAP+ reactive astrocytes [161], in addition to fibroblast and macrophages [160].

#### 6.1.4. Glial Bridge

Under normal conditions, Schwann cells form the myelin that surrounds and protects peripheral axons in mammals. Nevertheless, after PNS injury, these cells migrate from both nerve stumps to the injury site, forming a bridge that guides axonal crossing through the ablation gap during regeneration (Figure 2B). Similarly, after R model animal CNS injury, such as SCI, radial-glial cells present in the ependymal zone of the spinal cord form a glial bridge (Figure 2A). And, it has been described that some guidance cues participate in this bridge formation, as well as in the steering and maintenance of axonal traveling [138,162].

The mechanisms behind the glial bridge formation and maintenance, and of axonal traveling through the glial bridge are not completely elucidated. Ephrin/Eph signaling has been shown to play an important role during the formation (Figure 2B). On one hand, signaling between EphB2+ Schwann cells and Ephrin-B2+ fibroblast at the injury site, mediate the clustering of Schwann cells [163]. On the other hand, early after injury, EphA4 is upregulated in Schwann cells, inhibiting differentiation, and increasing proliferation, which is needed for preventing myelination and improving migration. After this early response, EphA4 is downregulated, coincident with differentiation and new axons myelination [164]. The maintenance of the nerve bridge morphology is in part due to peripheral macrophages expressing the repulsive cue Slit3 (Figure 2B). These macrophages localize surrounding the nerve gap, and restrain Robo1+ Schwann cells, fibroblasts, and regenerative axons in the injury site, impeding incorrect axonal traveling and cell migration outside the nerve path [165,166].

In regenerative animals, there is little evidence about a significant role of guidance cues during nerve bridge formation and/or maintenance. On one hand, in zebrafish PNS, it is suggested that Schwann cells steer axonal regeneration of DCC+ motor neurons by Netrin-1 secretion [167] and impede axon traveling through inappropriate trajectories by Slit1a repulsive signaling [168]. Interestingly, in the CNS of zebrafish, specifically in the spinal cord, it has been proposed the formation of a glial bridge after SCI, because axonal regeneration correlates with spinal cord radial-glial cells proliferation and migration to the injury site. These cells acquire a bipolar morphology forming a bridge between the rostral and caudal injury stump, allowing axon regeneration (Figure 2A) [169]. It could be suggested that Sema4D is involved in some processes related to the zebrafish CNS glial bridge because it is upregulated in regenerative motor neurons after SCI, and is associated with axonal regrowth and locomotory recovery [170]. Additionally, it has been suggested that the formation and maintenance of the glial bridge is controlled by the transcription factor Sox2 [163,166]. Interestingly, when comparing R versus NR *Xenopus leavis* stages, R-stages present radial glial morphology Sox2+ cells lining the spinal cord central canal [171], which proliferate and migrate in response to SCI [71,172]. Consistently, a low number of Sox2+ cells are detected in NR-stages [171], and they show a lesser and delayed proliferative response to SCI [71]. The glial bridge has not been described in this model organism; however, the above-mentioned data suggests that this process could occur in the spinal cord of R- but not in NR-stages.

### 6.2. Differential Expression of Guidance Cues after Nervous System Damage

Nervous system damage is accompanied by changes in guidance cues expression. Studying this differential expression and comparing it to models with better regenerative capacities, such as PNS and R animals, could help to explain the differences in regenerative capacities. In the present section, we will give an overview of what is known on guidance cues expression after SCI, PNS injuries, and after nervous system injuries in R animal models.

#### 6.2.1. Differential Expression of Guidance Cues after Different SCI Paradigms

After SCI, an upregulation of Sema3A levels is observed after adult rat spinal cord contusion [146], transection [146,148,173], and intraspinal motoneuron axotomy [174], mainly in the lesion center. In contrast to these observations, Hashimoto et al. (2004) detected a downregulation in Sema3A by semi-quantitative PCR. However, they showed that the decrease of Sema3A was due to neuronal death, and there was actually an upregulation in surviving neurons [173]. Therefore, it is important to consider the model, injury paradigm, cell type, and processes that are occurring. Additionally, Sema4F and the Neuropilin-2 receptor are upregulated in the affected motoneurons [174], and Sema3F is upregulated in the caudal portion of the transected spinal cord from *Xenopus* NR stages [69]. All the evidence suggests a prominent role of Sema in inhibiting spinal cord regeneration, although variations have been seen depending on the subtype and damage paradigm, such as the observation of downregulation of Sema4B and PlexinA3 in the caudal portion of the transected spinal cord from *Xenopus* NR stages [69] (Table 1). Additionally, in the intact spinal cord, Netrin-1 is continuously expressed in both neurons and oligodendrocytes [175,176], central canal cells, and meningeal cells [176]. Different SCI paradigms, including mouse spinal cord dorsal hemisection [160], rat sagittal myelotomy [177], and Allen’s spinal cord punch in rats [178], have shown an early, but transient Netrin upregulation, followed by downregulation. After SCI, in one study, the cells positive for Netrin-1 were identified as neurons and oligodendrocytes [177], while in another study it was detected in microglia/macrophages in the injury site, although in a more advanced stage after injury [160]. This contradictory evidence may be due to the temporarily different Netrin-1 response. In the case of Netrin receptors, most evidence shows a downregulation after SCI. UNC5A downregulation has been observed after sagittal myelotomy in rat spinal cord [177], dorsal hemisection [179], and transection [176]. While DCC presents an early upregulation, followed by downregulation after rat sagittal myelotomy [177], and an early downregulation is observed in *Xenopus*, [69] (Table 1). In the case of Slits, Slit2 expression in the adult spinal cord is low [180]. However, studies analyzing the first days after rat spinal cord weight-drop contusion show Slit2 downregulation, and upregulation of the Slit receptor Robo-1 in surviving neurons [181]. Studies focusing at later times after SCI show an upregulation of Slit2 [180,181], and no changes in Robo-1 expression [180]. Additionally, it seems that other components of the Slit family play different roles, as Slit1 and Slit3 are described to be upregulated after mouse spinal cord dorsal hemisection, where it has been detected in cells at the center of the lesion, including macrophages and fibroblasts [160] (Table 1). Finally, in the intact adult spinal cord, EphrinB2 and EphB2 are expressed by astrocytes and meningeal fibroblasts, respectively [151]. In adult rats, protein and mRNA levels of EphinB2, [151,182] and mRNA levels of EphrinA1 [183], show an initial downregulation, followed by a subsequent upregulation observed mainly in reactive astrocytes. Accordingly, studies focusing on the expression pattern of Ephrin receptors also showed an upregulation after injury. In rat spinal cord, EphB3 mRNA levels are increased after contusion [184], and both mRNA and protein levels after incision [185], while EphB2 mRNA and protein levels are also increased after SCI, both in the injury site, as well as in the surrounding tissues, principally in meningeal fibroblast [151,152]. For EphA4, the expression pattern after hemisection injury shows an increase [131], while after adult rat spinal contusion EphA4 mRNA levels show a biphasic behavior, with an initial downregulation, followed by an upregulation [186] (Table 1). A recent study analyzing guidance cues expression after mouse spinal cord crush has presented opposite results to what we have been previously discussed. However, this study dissected the expression of astrocytes and non-astrocytes cells in the injury gap [142], demonstrating the importance of studying every cell type in the different contexts as independent actors and as a whole, in order to have a full scenario of what is happening.

#### 6.2.2. Differential Expression of Guidance Cues after Different PNS Injury Paradigms

In PNS, Sema3A is upregulated in motor neurons after rat sciatic nerve transection and crush, and in femoral nerve transection [187,188]. Sema3F is upregulated in fibroblast after rat sciatic nerve transection, crush, and contusion [189,190], and Neuropilin-1 and Neuropilin-2 are upregulated in neurons and Schwann cells [189]. On the contrary, both Sema3A and Sema3F are downregulated in DRG neurons after rat DRG transection [189] (Table 2). In the case of the netrin family, it is normally upregulated after injury. Netrin-1 is upregulated mainly in Schwann cells [191,192] and macrophages [192], after sciatic nerve transection [191] and crush [193], and in an experimental inflammatory scenario of a rat model with a T-cell mediated autoimmune disease that affects PNS function [192]. Additionally, the Netrin receptor DCC, which mediates a chemoattractant response, presents an upregulation after adult rat sciatic nerve crush [193], and transection, localizing in some regenerative axons, but principally in recruited Schwann cells [194]. UNC5B is downregulated in the injury site after sciatic nerve transection [194], followed by an upregulation in the distal segment of the transected median nerve [195] (Table 2). In the case of the Slit family, it has been observed that Slit2 is upregulated after rat sciatic nerve transection, but not after crush, localizing in Schwann cells [196]. Slit1 is upregulated after sciatic nerve transection [197] and crush [198], while no changes are detected after dorsal rhizotomy [197]. In DRG cells, a biphasic expression is reported for Slit1, 2, and 3 after adult mouse sciatic nerve transection, with an initial downregulation, followed by a subsequent upregulation [165]. Slits receptors show similar expression patterns after injury, Robo1 and 2 were found to be transiently downregulated in mouse DRG cells after sciatic nerve transection [165], and upregulated in the rat sciatic nerve after transection [196,197]. Other molecules associated with Slit signaling, as the Slit-Robo GAPs (srGAPs) or the co-receptor Glypican-1, are also differentially expressed after sciatic nerve injury. srGAP1 and srGAP3(srGAP1/3) are upregulated in the adult mice DRG after nerve transection, in neurons and astrocytes [199], while Glypican-1 was also upregulated in DRG cells after nerve transection, dorsal column transection, and dorsal rhizotomy [200] (Table 2). Finally, EphA4 is upregulated after sciatic nerve crush, followed by a drastic downregulation [164] (Table 2). It has been described that EphA4 increases Schwann cells proliferation while inhibiting differentiation and thus, decreases myelination [164]. Therefore, the initially observed upregulation of EphA4 levels could be related to a proliferative process of Schwann cells after peripheral injury, and the following downregulation with differentiation and myelinization.

#### 6.2.3. Differential Expression of Guidance Cues after Nervous System Injuries in Regenerative Animal Models

The expression and contribution of guidance cues in regenerative animal models have not been extensively evaluated. Studies carried out in spinal cord transected lamprey, animal model which presents an incomplete axon regeneration [201], show an upregulation in Sema3 levels in microglia/macrophages in the injury site, while decreased levels of Sema4 and Netrin from dorsal cells and neurons close to the injury, respectively [201]. Additionally, a study comparing the transcriptomic response after spinal cord transection in *Xenopus laevis* R- and NR- stages, shows upregulation of Sema4B and PlexinA3 in R-stages, while a downregulation in NR-stages, and an upregulation of Sema3F in NR-stages [69] (Table 3). Netrin [201] and UNC5 [202] mRNA present an early downregulation near the lesion site after lamprey spinal cord transection, followed by an upregulation of UNC5 in poor regenerating neurons, an observation that was absent from the regenerative ones [202]. While the DCC receptor is slightly upregulated in *Xenopus* R-stages and downregulated in NR-stages, several Ephrin molecules tend to be slightly upregulated in R-stages while principally downregulated in NR-stages [69] (Table 1 and Table 3). In addition, Slit1 is upregulated after zebrafish peripheral motor nerve transection, specifically ventral and ventrolateral to the injury [168].

#### 6.2.4. Differential Expression of Guidance Cues Detected by Functional Studies

Leaving aside experiments analyzing the different guidance cues expression levels, there are a few gain- and loss- of function experiments analyzing the function of the different guidance cues in axon regeneration. Sema3D, and the receptor PlexinA1, are downregulated in KO:PTEN mice, which are able to regenerate the optical nerve after crush [203], while another study shows how transgenic mice with mutant PlexinA3 and PlexinA4 showed no regenerative improvements after mice SCI [204]. In goldfish, Netrin-1 was found binding to RGC regenerative axons, suggesting that these neurons expressing Netrin receptors had a possible role in this signaling during regeneration [205]. Even more, the Planarian homologs of Netrin and DCC were found to be essential for proper CNS regeneration, as RNAi mediated KD, impairs the correct neuronal patterning [206]. While Neogenin, the Netrin receptor involved in the repulsive response, is expressed mainly in poor regenerative neurons of the adult lamprey spinal cord, and its levels do not variate after injury. Nevertheless, KD:Neogenin animals show improvement of reticulospinal neurons regeneration [207]. In 2007, Cebrià and collaborators detected a Slit ortholog in planaria (Smed-Slit), which was found to be expressed after head amputation, and knocking down this cue causes a failure in the regeneration patterning, resulting in nervous system regeneration at the midline instead of bi-laterally [208], while Smed-Slit regulation by the microRNAs family, miR-124 is important for proper brain and eyes regeneration as well [209]. Transgenic mice lacking EphrinB2 [153] and EphrinB3 [210] show improved locomotor recovery after SCI, supporting the notion of an anti-regenerative role for this cue. However, blocking EphrinA1 expression with antisense oligonucleotides shows the opposite effect on locomotor recovery [183], suggesting a pro-regenerative role for this ligand. Finally, in the mammal regenerative opossum, an upregulation of EphB4 was observed, coincident with the intrinsic loss of spinal cord regenerative capacity during development advance [211]. All the aforementioned evidence supports the role of guidance cues after nervous system injury and regeneration, although more studies are needed in order to set up an expression pattern and a possible mechanism by which they modulate the regenerative process.

**Table 1 ijms-22-08344-t001:** Differential expression of guidance cues after different SCI paradigms.

Guidance Cues	Injury	Expression Levels	mRNA/Protein	Comments	Ref.
Sema3A	Contusion, transection and motoneurons axotomy, in rats	Upregulated	mRNA/protein	In fibroblast and motoneurons in the lesion center	[149,151,173,174]
Sema3F	*Xenopus* spinal cord transection	Upregulated	mRNA	Caudal portion of NR stages spinal cord	[69]
Sema4F/ Neuropilin2	Rat intraspinal motoneurons axotomy	Upregulated	mRNA	In affected motoneurons	[174]
Sema4B/ PlexinA3	*Xenopus* spinal cord transection	Downregulated	mRNA	Caudal portion of NR stages spinal cord	[69]
Netrin-1	Mouse dorsal hemisection, rat sagittal myelotomy, and rat spinal cord punch	Upregulated/ Downregulated	mRNA/protein	Upregulation in activated microglia and macrophages in the lesion epicenter. Detected in neurons and oligodendrocytes from sparse tissue.	[160,177,178]
DCC	Rat sagittal myelotomy and *Xenopus* spinal cord transection	Upregulated/ Downregulated	mRNA/protein	Early and transient upregulation, followed by downregulation in lesion epicenter	[69,177]
UNC5	Rat sagittal myelotomy, spinal cord dorsal hemisection, and transection	Downregulated	mRNA/protein	Downregulation of UNC5A-D in neurons and oligodendrocytes	[176,177,179]
Slit-1/3	Mouse dorsal hemisection	Upregulated	mRNA	In macrophages and fibroblasts in the lesion epicenter	[160]
Slit-2	Rat spinal cord punch and contusion	Upregulated	mRNA/protein	Early and transient upregulation in neurons	[178,180]
Robo-1	Rat spinal cord contusion	Upregulated	mRNA/protein	In surviving neurons	[181]
Ephrin-B2	Thoracic spinal cord transection, and dorsal hemisection in rat	Upregulated	mRNA/protein	Mainly in reactive astrocytes in the glial scar. One study shows an early and transient downregulation	[134,154,182]
EphrinB3	Rat spinal cord crush and dorsal hemisection	Down- and upregulated	mRNA/protein	Downregulation is most probably due to cell death. And upregulation in astrocytes	[134,182]
EphrinB1	Rat spinal cord dorsal hemisection	Upregulated	Protein	In astrocytes	[134]
EphrinA1	Rat spinal cord contusion	Downregulated/ Upregulated	mRNA/protein	Expressed in reactive astrocytes and neurons	[183]
EphA4	Rat spinal cord dorsal hemisection	Upregulated	mRNA/protein	In axons rostrally, and astrocytes in the injury site	[134,182]
	Rat spinal cord contusion	Downregulated/ Upregulated	mRNA/protein	In neurons caudal to injury and astrocytes rostral and caudal to the lesion epicenter	[186,212]
EphB3	Rat spinal cord contusion and transection	Upregulated	mRNA/protein	In astrocytes in the epicenter and in neurons rostral and caudal to the injury site	[184,185]
EphB2	Spinal cord transection and contusion	Downregulation/ Upregulation	mRNA/protein	Expressed in activated astrocytes and in fibroblasts invading the lesion site	[154,155]

Downregulation/Upregulation: Early and transient downregulation, followed by upregulation; Down- and upregulated: some studies show downregulation and others show upregulation.

**Table 2 ijms-22-08344-t002:** Differential expression of guidance cues after different PNS injury paradigms.

Guidance Cues	Injury	Expression Levels	mRNA/Protein	Comments	Ref.
Sema3A	Rat sciatic nerve transection and crush, and femoral nerve transection	Upregulated	mRNA	In ipsilateral spinal cord motor neurons	[187,188]
	Rat dorsal root transection	Downregulated	mRNA	In DRG neurons	[189]
Sema3F	Rat sciatic nerve transection, crush, and contusion	Upregulated	mRNA	In epineurial fibroblast and perineurium	[189,190]
	Rat DRG transection	Downregulated	mRNA	In neurons	[189]
Neuropilin-1/2	Rat DRG and sciatic nerve transection	Upregulated	mRNA	DRG neurons	[189]
Neuropilin-2	Rat sciatic nerve transection	Upregulated	mRNA/protein	In neuron caudal to injury, and in Schwann cells in the perineurium and epineurium	[190]
Netrin-1	Rat sciatic nerve transection and crush, and experimental autoimmune neuritis	Upregulated	mRNA/protein	In Schwann cells, macrophages, ECM, and some axons	[191,192,193]
DCC	Rat sciatic nerve crush and transection	Upregulated	mRNA/protein	In DRG neurons, and Schwann cells	[193,194]
UNC5B	Rat sciatic nerve, and mice median nerve transection	Downregulated/ Upregulated	mRNA/protein	Downregulation in DRG neurons, followed by an increase in the distal nerve segment	[194,195]
EphA4	Rat sciatic nerve crush	Upregulated/ Downregulated	protein	In Schwann cells at the injury site	[164]
Slit1	Rat and mouse sciatic nerve transection and rat sciatic nerve crush	Down- and upregulated	mRNA/protein	Downregulated in mouse DRG neurons, and upregulated in rat DRG neurons and non-neuronal cells	[165,197,198]
Slit2	Rat and mouse sciatic nerve transection	Down- and upregulated	mRNA	Downregulated in DRG neurons and in the injury site, and upregulated in Schwann cells and proximal stump	[165,196]
Slit3	Mouse sciatic nerve transection	Down- and upregulated	mRNA	Downregulated in neurons and upregulated in macrophages surrounding the nerve bridge	[195]
Robo1	Mouse and rat sciatic nerve transection	Downregulated/ Upregulated	mRNA	Early downregulation in DRG neurons, and upregulation in Schwann cells in the distal stump and in the nerve bridge	[165,196]
Robo 2	Mouse and rat sciatic nerve transection	Down- and upregulated	mRNA/protein	In mouse a transient downregulation and in rats a transient upregulation	[165,196,197]
srGAP1/3	Mouse sciatic nerve transection	Upregulated	mRNA/protein	In the ipsilateral of DRG neurons	[199]
Glypican-1	Rat sciatic nerve transection	Upregulated	mRNA/protein	In the ipsilateral of DRG neurons and in neighboring non-neuronal cells	[200]

Downregulation/Upregulation: Early and transient downregulation, followed by upregulation; Down- and upregulated: some studies show downregulation and others show upregulation.

**Table 3 ijms-22-08344-t003:** Differential expression of guidance cues after nervous system injuries in regenerative animal models.

Guidance Cues	Injury	Expression Levels	mRNA/Protein	Comments	Ref.
Sema3	Larval lampreys spinal cord transection	Upregulated	mRNA	In microglia/macrophages and in some neurons near the lesion site	[201]
Sema4	Larval lampreys spinal cord transection	Downregulated	mRNA	Downregulated in dorsal cells, and present rostral and caudal to injury, but absent from scar	[201]
Sema4B/ PlexinA3	*Xenopus* spinal cord transection	Upregulated	mRNA	The caudal portion of R stages spinal cord	[69]
Netrin	Larval lampreys spinal cord transection	Downregulated	mRNA	In neurons close to the injury site	[201]
DCC	*Xenopus* spinal cord transection	Upregulated	mRNA	The caudal portion of R stages spinal cord	[69]
UNC-5	Larval lampreys spinal cord transection	Downregulated/ Upregulated	mRNA	Upregulation in neurons with poor regenerative capacity, and absent from the ones with good capacity	[202]
Ephrin	*Xenopus* spinal cord transection	Upregulated	mRNA	The caudal portion of R stages spinal cord	[69]
Slit1	Zebrafish peripheral motor nerves transection	Upregulated	mRNA	Ventral and ventrolateral to the injury	[168]

Downregulation/Upregulation: Early and transient downregulation, followed by upregulation.

## 7. Treatments: Guidance Cues Regulation

A great effort has been generated on elucidating new therapeutic targets and new therapeutic approaches, for improving axon regeneration after injuries and diseases. One used methodology consists of the inhibition of guidance cues, either by pharmacological inhibitors, directed antibodies, or microRNA. Sema3A inhibition has been analyzed as a treatment for different CNS injuries, showing promising results. For example, studies using anti-Sema3A antibodies show an increase in RGCs density in rat axotomized optic nerve [213], and improvements in axonal traveling at the injury site of the transected inferior alveolar nerve [214]. In a similar way, the usage of different Sema3A inhibitors has shown improvement in the recovery after middle cerebral artery occlusion [149] and SCI [148,215]. For the autoimmune disease scenario of multiple sclerosis, inhibition of Sema4D with antibodies showed a decrease in the severity of the experimental rodent model EAE [216]; even more, this antibody reached randomized phase 1 trial for adult patients with multiple sclerosis, showing safety and tolerability for the individuals, and favoring the interest to continue studying this methodology as a therapeutic treatment [217]. Another inhibitory approach consists of the administration of recombinant peptides that antagonize specific receptors. This approach has been evaluated for EphA4 after SCI [132,218,219], showing an improvement in axonal regeneration and functional/locomotory recovery. Finally, an increasing number of studies have used RNAi or antisense oligonucleotides for impeding guidance cue mRNA translation. Pro-regenerative effects in the injured spinal cord have been observed when targeting EphrinA3 [220], EphrinB2 [154], EphB2 [152], EphA4 [221], and in DRG cells when targeting EphrinA3 [222], while disadvantageous effects in locomotor recovery after SCI were seen after targeting EphrinA1 with anti-sense oligonucleotides [183], all these studies have provided important information for the elucidation of new therapeutic targets, and should also be used for other guidance cues family members. Recombinant peptides have also been used for potentiation of suggested pro-regenerative guidance cues, as Netrin-1. In this regard, administration of Netrin-1 recombinant peptides in the injury site has shown promising results when used on injured brains, as after hemorrhage [124,223] and after middle cerebral artery occlusion [125]. However, in the peripheral sciatic nerve, Netrin-1 administration shows no effect [194]. This approach can be used as well for preventive treatments, as administration of recombinant Slit2 before adult rat surgical brain injury results in the reduction of neuroinflammation [134], and BBB permeability, by increasing junction proteins expressions [135].

Another methodology that has gained a lot of interest is nerve or hydrogel grafting in the injury site. Grafting peripheral nerve extracts, commonly from the sciatic nerve, into injured central nerves, is due to the better regenerative capacity of peripheral nerves in comparison to central nerves [224]. An interesting approach shows that in injured rat optic nerve, the Netrin-1 signal appears transiently after PN grafting [205], while DCC and UNC5B mRNA levels were similarly downregulated after injury with or without PN grafting [225]. Additionally, PN grafting in the transected optic nerve led to the maintenance of the EphA5 nasal-temporal gradient in RGCs, and to the establishment of a rostro-caudal EphrinA2 gradient in the superior colliculus [226], supporting the association of guidance cues with the efficacy of this type of treatment. Different grafting approaches have been implemented in order to evaluate if guidance cues improve neurons regeneration, as genetically modified fibroblasts overexpressing Netrin-1 grafting in the injured rat spinal cord where regeneration gets worse [176], or a Sema3A gradient in a hydrogel-based device grafted in the injured adult rat cortical brain, improves the neuronal progenitors’ migration and differentiation [72]. More studies are needed in order to correctly elucidate the pro-regenerative guidance cues to properly exploit the grafting methodology.

## 8. Conclusions

It is plausible that during the last decades much has been elucidated about guidance cues function during axon development. The firstly determined components included the guidance cues family members, if they are secreted or membrane-bound, if they act at long or short distances, and the signaling cascades they activate, among others. More recently, evidence shows how these guidance cues regulate the growth cone cytoskeleton dynamics and crosstalk, resulting in filopodia, lamellipodia, contact points, and exploring MT regulation. Although what has drawn the most attention is how so few canonical guidance cues exist; they manage to steer the growth cone advance in so many complex environments, allowing axons to express the same guidance cues and/or receptors, or axons traveling together, to respond differently. The evidence shown herein demonstrates the complex crosstalk between the guidance cues, showing that every group of axons enervating different targets need to be studied separately and that an integrative approach needs to be used in order to elucidate how to improve axon regenerative capacities.

## Figures and Tables

**Figure 1 ijms-22-08344-f001:**
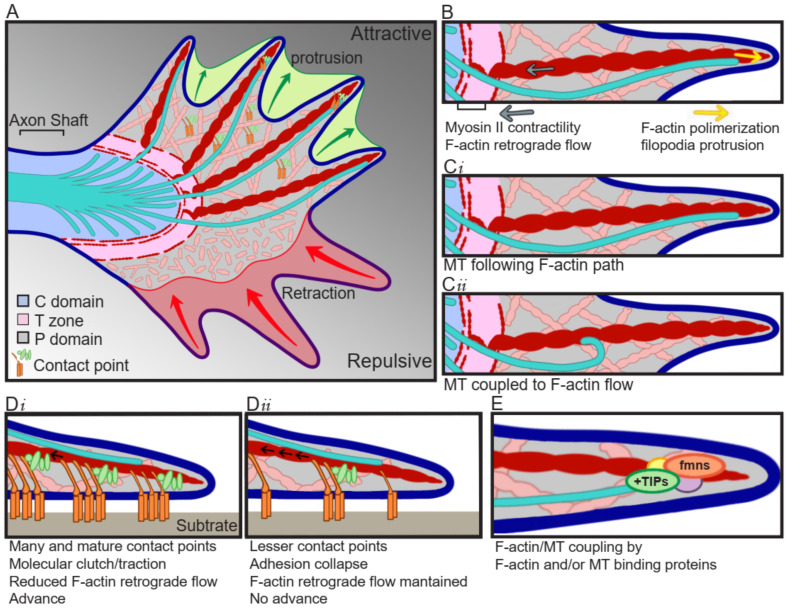
The growth cone. (**A**). Schematic representation of the axonal growth cone structure and cytoskeletal response to attractive and repulsive guidance cues. The axonal growth cone is composed of three principal domains: the central (**C**) domain (light blue), where the microtubules (cyan) coming from the axon shaft are splayed; the transition (T) zone (pink), where contractile F-actin arcs (red) are organized in a semi-circle structure separating the C domain from the peripheral (P) domain (gray), which is composed by filopodia (F-actin bundles, red) and lamellipodia (F-actin mesh, peach). In the upper right, an attractive gradient is represented, towards which the growth cone moves (green arrows). New lamellipodia (light green) and contact points are formed towards the leading edge, while in the opposite direction, F-actin is disassembled, and filopodia and lamellipodia retract (red arrows). (**B**). Schematic representation of F-actin dynamics in the growth cone. F-actin bundles polymerizing towards the leading edge of the growth cone (yellow arrow), generating growth cone protrusive forces, while the F-actin is subjected in turn to a retrograde movement towards the T zone, driven by myosin II (gray arrow). (**C**). Schematic representation of MT dynamics in the growth cone. An exploring MT enters the P domain following the F-actin path (**C*i***), but at the same time it, couples to F-actin and thus, to F-actin retrograde flow (**C*ii***). (**D**). Schematic representation of molecular clutch formation. Recruitment of many integrin receptors (orange) result in adhesive molecules binding (green), and reduced F-actin retrograde flow and growth cone advance (**D*i***), while few integrin receptors result in adhesion collapse, and no advance is observed (**D*ii***). (**E**). Schematic representation of F-actin and MT binding proteins involved in F-actin/MT coupling in growth cone filopodia.

**Figure 2 ijms-22-08344-f002:**
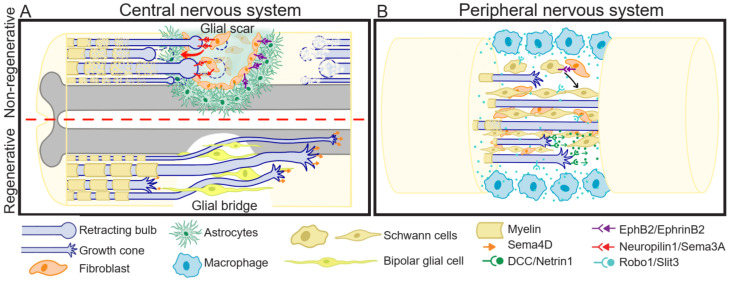
Guidance cues are associated with the glial scar and glial bridge. (**A**). Schematic representation of non-regenerative (NR) and regenerative (R) spinal cord response after injury. Injured axons from NR animal models (upper panel) fail to form a growth cone but instead form a retracting bulb guiding axonal degeneration, while a glial scar is formed in the injury site, which expresses mainly repulsive guidance cues. Injured axons from R animal models (lower panel) generate a growth cone, which travels through the ablation gap following a glial bridge formed by bipolar glial cells, that also expresses guidance cues. (**B**). Schematic representation of a peripheral nervous system axon after injury. Damaged axons can regenerate by extending through the ablation gap, following a glial bridge formed by Schwann cells and macrophages, which express different guidance cues that restrict a corridor for axons to grow.

## Data Availability

Not applicable.

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
