# Peer review of "Unraveling Axon Guidance during Axotomy and Regeneration"

_ijms, 2021, doi:10.3390/ijms22158344_

Round 1

Reviewer 1 Report

Title

Title is appropriate because it is completely informative about the contents of the paper.

Abstract

The abstract respects the rules of the journal. The background and the aim are interesting. In the design is present the type of study.

Text

The introduction of the study clearly sums up the aim of the study. The authors provide a full view for performing the study based on a review of the scientific literature. Furthermore, they define well the aim of the manuscript.

The study was better structured and carried out. The various topics in the paragraphs are reported clearly, but they should be written more fluently, as they are redundant in some sentences.

To eliminate in the text ” For a comprehensive review of this topic see”,  the reference number is already present, that is enough.

References

They are qualified and integrated with the aim. The reference list follows the format for the journal.

Figures

They highlight the key points and they are good quality.

Tables

They sum up the study concisely and clearly, there is no match between the groups in tables and in text.

General comments

The purpose of the study is original and the study has been fixed correctly; but the various topics in the paragraphs are reported clearly, but they should be written more fluently, as they are redundant in some sentences.

Author Response

We thank you for your very helpful suggestions to improve the manuscript.  We have addressed all the comments.  Below are the reviews and our response to them (in blue).

Title: Title is appropriate because it is completely informative about the contents of the paper.

Abstract: The abstract respects the rules of the journal. The background and the aim are interesting. In the design is present the type of study.

Text: The introduction of the study clearly sums up the aim of the study. The authors provide a full view for performing the study based on a review of the scientific literature. Furthermore, they define well the aim of the manuscript.

The study was better structured and carried out. The various topics in the paragraphs are reported clearly, but they should be written more fluently, as they are redundant in some sentences. We added some phrases and/or paragraphs to string together the different topics and make the manuscript more fluently.

To eliminate in the text ” For a comprehensive review of this topic see”,  the reference number is already present, that is enough. We eliminated in the text ” For a comprehensive review of this topic see”.

References: They are qualified and integrated with the aim. The reference list follows the format for the journal.

Figures: They highlight the key points and they are good quality.

Tables: They sum up the study concisely and clearly, there is no match between the groups in tables and in text. We added sub-headings to match the tables to the text.

General comments: The purpose of the study is original and the study has been fixed correctly; but the various topics in the paragraphs are reported clearly, but they should be written more fluently, as they are redundant in some sentences. We made changes to make the manuscript more fluently and also made an English spell check.

Reviewer 2 Report

Domínguez-Romero and Slater review recent findings about axon guidance during axotomy and regeneration. Figures 1 and 2 represent schematic diagrams of growth cone and guidance cues in the CNS and PNS. Tables 1-3 demonstrate a variety of guidance cue ligands and receptors. The authors should explain as much technical terms as possible so that this manuscript can be understood by non-specialists. There are so many points that may serve to amend this manuscript, as follows:

Major points:

  1. Many abbreviations are used in this manuscript. It would be nice to give a list of abbreviations to help the reader understand this manuscript.
  2. Figure 1 should be placed near the fourth paragraph on page 2.
  3. Tables 1-3: please check once again if the phenomena shown in these tables are indeed mentioned in the text. “Macromolecule” word seems inappropriate. Please use another word.
  4. There is a lot of grammatically incorrect English in this manuscript. For example, line 62: not “question” but “questions”; line 117: not “where” but “which”?; line 122: not “are” but “is”; line 123: not “depends” but “depend”; line 145: not “define” but “defines”; line 177: not “where” but “which”?; line 178: not “are” but “is”; line 245: not “are” but “is”; line 253: not “where” but “which”?; line 261: not “suggest” but “suggests”?; line 264: not “are” but “is”; line 272: not “mediated” but “mediate”; line 279: not “analyzes” but “analyses”?; line 325: not “on” but “of”; line 349: not “intermediates TCA” but “intermediate TCA”?; not “are” but “is”; line 350: not “kind” but “kinds”?; line 452: not “lead” but “leads”; line 455: not “regulate” but “regulates”; line 474: not “correspond” but “corresponds”?; line 558: not “show” but “shows”; line 560: not “present” but “presents”; line 567: please correct “a different roles”; line 572: not “initially” but “initial”?; line 597: not “present” but “presents”; line 616: not “initial” but “initially”?; line 622: not “present” but “presents”; line 626: not “show” but “shows”; line 640: not “show” but “shows”; line 649: not “Planaria” but “planaria”; line 678: not “is” but “are”; line 693: not “show” but “shows”; line 694: not “present” but “presents”; line 695: not “cause” but “causes”; line 698: not “present” but “presents”; not “induce” but “induces”; line 706: not “support” but “supports”; line 723: not “cause” but “causes”; line 728: not “generate” but “generates”; line 729: not “show” but “shows”; line 748: not “aggregate” but “aggregates”; line 786: not “enhance” but “enhances”; line 787: not “analyzes” but “analyses”; line 797: not “have” but “has”; line 802: not “surround” but “surrounds”; not “protect” but “protects”; line 825: not “where” but “which”?; line 826: not “impede” but “impedes”; line 830: not “where” but “which”?; line 842: not “occurs” but “occur”; line 848: not “express” but “expresses”; line 849: not “travel” but “travels”; line 852: not “express” but “expresses”; line 883: not “show” but “shows”; line 890: not “show” but “shows”; line 891: not “appear” but “appears”; line 896: not “have” but “has”; line 905: not “first” but “firstly”. There appear to be more English mistakes than pointed out above. Please check your manuscript very carefully.

Specific points:

  1. 1A: not “Atractive” but “Attractive”.
  2. Line 93: not “Fig.1A” but “Fig. 1A”.
  3. Line 106: “ECM” should be defined.
  4. Line 184: not “Integrin” but “integrin”.
  5. Line 202: please expand “VASP”.
  6. Does “Mical” in lines 208 and 213 mean “MICAL” in line 446? Please reply to this question.
  7. Line 224: does “KD” mean knockdown? If so, “KD” should be defined in this line but not in line 709.
  8. Line 225: not “Hamster” but “hamster”.
  9. Line 232: “APC” should be defined.
  10. Line 234: “CLASP” should be defined.
  11. Line 236: “XMAP215” should be defined.
  12. Line 237: “TACC” should be defined.
  13. Line 256: “DAAM” should be defined. Is this “DAAM” the same as the “Daam” on line 271? Please make this point clear.
  14. Line 257: “EB1” should be defined.
  15. Line 267: “CLIP-170” should be defined.
  16. Line 271: “INF” should be defined.
  17. Line 340: not “tyrosine-kinase receptors” but “receptor tyrosine-kinases”?
  18. Line 345: is “de receptor” OK?
  19. Line 348: not “thalamocortical (TCA)” but “thalamocortical area (TCA)”?
  20. Lines 357 and 358: please check whether the English is right.
  21. Line 372: please define “SCI” in this line but not in line 487.
  22. Line 374: please explain “dpi” shortly.
  23. Line 420: please expand “CS”.
  24. Line 453: please expand “CRMP”.
  25. Lines 465 and 466: both “Recent” and “recently” should not be used.
  26. Lines 481 and 482: please check whether the English is right.
  27. Lines 489 and 490: please check whether the English is right.
  28. Line 500: not “… regenerate, interestingly …” but “… regenerate; interestingly …”.
  29. Line 503: not “non regenerative” but “non-regenerative”.
  30. Lines 532 and 533: not “… what it is known on …” but “… what is known on …”?
  31. Line 549: not “both,” but “both”.
  32. Line 558: “Unc5” or “UNC5” (Table 1) should be used.
  33. Line 570: not “during” but “in”?
  34. Line 570: please explain shortly a difference between “EphrinB2” and “EphB2”.
  35. Line 592: please correct the English in this line.
  36. Line 596: not “PNs” but “PNS”.
  37. Line 609: do “srGAPs” mean “srGAP1/3” in Table 2? If so, please make this point clear.
  38. The second paragraph on page 13: not “Lamprey” but “lamprey”; not “Zebrafish” but “zebrafish”.
  39. Line 667: not “an” but “and”?
  40. Line 695: NO should be defined in this line but not in line 696.
  41. Line 697: not “Lipopolysaccharide” but “lipopolysaccharide”.
  42. Lines 721 and 722: not “Tumor Necrosis Factor” but “tumor necrosis factor”; not “Interferon” but “interferon”.
  43. Line 788: not “Glial” but “glial”.
  44. Line 837: not “non regenerative” but “non-regenerative”.
  45. Line 866: please expand “MS”; please correct “where”.

Author Response

We thank you for your very helpful suggestions to improve the manuscript.  We have addressed all the comments.  Below are the reviews and our response to them (in blue).

  1. Many abbreviations are used in this manuscript. It would be nice to give a list of abbreviations to help the reader understand this manuscript: a list of abbreviation was added at the end of the manuscript.
  2. Figure 1 should be placed near the fourth paragraph on page 2: we moved the figure to the beginning of page 3.

 Tables 1-3: please check once again if the phenomena shown in these tables are indeed mentioned in the text. “Macromolecule” word seems inappropriate. Please use another word: We changed Macromolecule for mRNA/protein

  1. There is a lot of grammatically incorrect English in this manuscript. For example, line 62: not “question” but “questions”; line 117: not “where” but “which”?; line 122: not “are” but “is”; line 123: not “depends” but “depend”; line 145: not “define” but “defines”; line 177: not “where” but “which”?; line 178: not “are” but “is”; line 245: not “are” but “is”; line 253: not “where” but “which”?; line 261: not “suggest” but “suggests”?; line 264: not “are” but “is”; line 272: not “mediated” but “mediate”; line 279: not “analyzes” but “analyses”?; line 325: not “on” but “of”; line 349: not “intermediates TCA” but “intermediate TCA”?; not “are” but “is”; line 350: not “kind” but “kinds”?; line 452: not “lead” but “leads”; line 455: not “regulate” but “regulates”; line 474: not “correspond” but “corresponds”?; line 558: not “show” but “shows”; line 560: not “present” but “presents”; line 567: please correct “a different roles”; line 572: not “initially” but “initial”?; line 597: not “present” but “presents”; line 616: not “initial” but “initially”?; line 622: not “present” but “presents”; line 626: not “show” but “shows”; line 640: not “show” but “shows”; line 649: not “Planaria” but “planaria”; line 678: not “is” but “are”; line 693: not “show” but “shows”; line 694: not “present” but “presents”; line 695: not “cause” but “causes”; line 698: not “present” but “presents”; not “induce” but “induces”; line 706: not “support” but “supports”; line 723: not “cause” but “causes”; line 728: not “generate” but “generates”; line 729: not “show” but “shows”; line 748: not “aggregate” but “aggregates”; line 786: not “enhance” but “enhances”; line 787: not “analyzes” but “analyses”; line 797: not “have” but “has”; line 802: not “surround” but “surrounds”; not “protect” but “protects”; line 825: not “where” but “which”?; line 826: not “impede” but “impedes”; line 830: not “where” but “which”?; line 842: not “occurs” but “occur”; line 848: not “express” but “expresses”; line 849: not “travel” but “travels”; line 852: not “express” but “expresses”; line 883: not “show” but “shows”; line 890: not “show” but “shows”; line 891: not “appear” but “appears”; line 896: not “have” but “has”; line 905: not “first” but “firstly”. There appear to be more English mistakes than pointed out above. Please check your manuscript very carefully: We changed all the mentioned grammatically English in the manuscript, and others that were found

Specific points:

  1. 1A: not “Atractive” but “Attractive”: modified
  2. Line 93: not “Fig.1A” but “Fig. 1A” : modified
  3. Line 106: “ECM” should be defined: defined
  4. Line 184: not “Integrin” but “integrin” : modified
  5. Line 202: please expand “VASP”: defined
  6. Does “Mical” in lines 208 and 213 mean “MICAL” in line 446? Please reply to this question: Yes, it refers to the same protein, we changed all of them to MICAL, and the definition was moved to the first time that the protein is mentioned
  7. Line 224: does “KD” mean knockdown? If so, “KD” should be defined in this line but not in line 709: changed
  8. Line 225: not “Hamster” but “hamster”: changed
  9. Line 232: “APC” should be defined: defined
  10. Line 234: “CLASP” should be defined: defined
  11. Line 236: “XMAP215” should be defined: defined
  12. Line 237: “TACC” should be defined: defined
  13. Line 256: “DAAM” should be defined. Is this “DAAM” the same as the “Daam” on line 271? Please make this point clear: Yes, is the same protein, we changed all to DAAM, and added the definition
  14. Line 257: “EB1” should be defined: defined
  15. Line 267: “CLIP-170” should be defined: defined
  16. Line 271: “INF” should be defined: defined
  17. Line 340: not “tyrosine-kinase receptors” but “receptor tyrosine-kinases”?:changed
  18. Line 345: is “de receptor” OK? :changed to the receptor
  19. Line 348: not “thalamocortical (TCA)” but “thalamocortical area (TCA)”? :changed
  20. Lines 357 and 358: please check whether the English is right: it was checked, and changed from When Neogenin binds Netrin-1 it elicits an attractive response, but when it binds the repulsive guidance molecule (RGM)mediates a repulsive response. To: When Neogenin binds Netrin-1, it elicits an attractive response, but when binds the repulsive guidance molecule (RGM), it mediates a repulsive response.
  21. Line 372: please define “SCI” in this line but not in line 487: changed
  22. Line 374: please explain “dpi” shortly: explained
  23. Line 420: please expand “CS”: expanded
  24. Line 453: please expand “CRMP”: defined
  25. Lines 465 and 466: both “Recent” and “recently” should not be used: removed
  26. Lines 481 and 482: please check whether the English is right: changed from: Most of what we know about guidance cues steering axonal growth, is based on studies during neuronal development, to: Most of what we know about the role of guidance cues in the steering of axonal growth, is based on studies during neuronal development.
  27. Lines 489 and 490: please check whether the English is right: changed from: SCI affects nerves involved in receiving sensory information and controlling motor response, generating paralysis beneath the injury site, while internal organs result disconnected from central nervous system regulation, to: SCI affects nerves involved in receiving sensory information and controlling motor response, generating paralysis beneath the injury site. Additionally, after SCI, some internal organs result disconnected from central nervous system regulation.
  28. Line 500: not “… regenerate, interestingly …” but “… regenerate; interestingly …”: changed
  29. Line 503: not “non regenerative” but “non-regenerative”: changed
  30. Lines 532 and 533: not “… what it is known on …” but “… what is known on …”?:changed
  31. Line 549: not “both,” but “both”: changed
  32. Line 558: “Unc5” or “UNC5” (Table 1) should be used: all Unc were changed by UNC
  33. Line 570: not “during” but “in”?: changed
  34. Line 570: please explain shortly a difference between “EphrinB2” and “EphB2”: the difference between Ephrin and Eph is explained in line 344
  35. Line 592: please correct the English in this line: changed from: In PNS, Sema3A is upregulated in motor neurons after rat sciatic nerve transection and crush, and in femoral nerve transection as well [128,129], while Sema3F in fibroblast after rat sciatic nerve transection, crush, and contusion [130,131], to: In PNS, Sema3A is upregulated in motor neurons after rat sciatic nerve transec-tion and crush, and in femoral nerve transection as well [128,129], while Sema3F is upregulated in fibroblast after rat sciatic nerve transection, crush, and contusion [130,131]
  36. Line 596: not “PNs” but “PNS”: changed
  37. Line 609: do “srGAPs” mean “srGAP1/3” in Table 2? If so, please make this point clear: cleared
  38. The second paragraph on page 13: not “Lamprey” but “lamprey”; not “Zebrafish” but “zebrafish”: changed
  39. Line 667: not “an” but “and”?: changed
  40. Line 695: NO should be defined in this line but not in line 696: changed
  41. Line 697: not “Lipopolysaccharide” but “lipopolysaccharide”: changed
  42. Lines 721 and 722: not “Tumor Necrosis Factor” but “tumor necrosis factor”; not “Interferon” but “interferon”: changed
  43. Line 788: not “Glial” but “glial”: changed
  44. Line 837: not “non regenerative” but “non-regenerative”: changed
  45. Line 866: please expand “MS”; please correct “where”: expanded and corrected

Reviewer 3 Report

The manuscript by Dominguez-Romero and Slater provides an extensive overview on the known cues that regulate growth cone guidance and pathfinding during development and upon axotomy, both in regenerative and non regenerative contexts. The work is very accurate and derives from an extensive revision of the literature as well as of more recent findings. However, some points need to be addressed before publication:

  • In general, sections are long and sometimes heavy to read. Sub-headings and additional figures may help appreciating in a more schematic and clear way the contribution of the single cues described in the text (e.g. Sections 5 and 6.1).
  • Colors in Figure 1 are a bit confusing, alternative colors may allow a better visualization of growth cone domains and components and their engagement in growth cone dynamics. For example in panels A-C contact points are in red, while in panels Di and Dii they are in green. Moreover, characters’ size in the legends is too small.
  • When referring to PNS regenerative capabilities, please specify that PNS axons can regenerate to a certain extent (in the abstract and throughout the text).
  • Refs 101 and 106 cited in Section 6 refer to studies performed in zebrafish, while in the text authors are also referring to mammals. Additional references are thus required.
  • This Reviewer believes that Section 5, which is dedicated to the cross talk among the different guidance cues orchestrating growth cone dynamics, and the contribution of different cell types in the process, will be more complete if additional molecules (e.g. chemokines) and/or signaling pathways that participate in axonal pathfinding during embryonic development, and in regeneration, e.g. the CXCL12-CXCR4 axis (Lieberham et al., Neuron 2005), will be included in the discussion.
  • When speaking about the difference in regeneration competence of CNS axons vs PNS ones, please include and comment the studies by Hellal et al., Science 2011; Tom et al., J Neurosci. 2004; Ertük et al., J Neurosci. 2007, which deal with the formation of retraction bulbs, and highlight the importance of microtubule stabilization of axons and growth cones (occurring in the PNS and not in the CNS) in order to counteract regeneration failure
  • Language editing required

Author Response

We thank you for your very helpful suggestions to improve the manuscript.  We have addressed all the comments.  Below are the reviews and our response to them (in blue).

  • In general, sections are long and sometimes heavy to read. Sub-headings and additional figures may help appreciating in a more schematic and clear way the contribution of the single cues described in the text (e.g. Sections 5 and 6.1). Sub-headings were added to sections 5. Section 6 was re-organized and new sub-headings were added. Additionally, some phrases and/or paragraphs were added to string together the different topics.
  • Colors in Figure 1 are a bit confusing, alternative colors may allow a better visualization of growth cone domains and components and their engagement in growth cone dynamics. For example in panels A-C contact points are in red, while in panels Di and Dii they are in green. Moreover, characters’ size in the legends is too small. Colors used for growth cone domains and contact point were changed, characters size in legends were augmented.
  • When referring to PNS regenerative capabilities, please specify that PNS axons can regenerate to a certain extent (in the abstract and throughout the text). In the introduction was added: After neuronal injuries in mammals, only axons in the peripheral nervous system (PNS) are able to regenerate to some extent. Which was further explained in section 6.1
  • Refs 101 and 106 cited in Section 6 refer to studies performed in zebrafish, while in the text authors are also referring to mammals. Additional references are thus required. Reference 101 refers mostly to zebrafish but it also has a section about mammals. We added Ertürk., et al. 2007 in point 2 of section 6.1 for more evidence on mammals. We also added Min., et al. 2021 and He., et al. 2020 in point 3 of section 6.1 for more evidence on glial bridge in PNS and glial scar in mammals, respectively.
  • This Reviewer believes that Section 5, which is dedicated to the cross talk among the different guidance cues orchestrating growth cone dynamics, and the contribution of different cell types in the process, will be more complete if additional molecules (e.g. chemokines) and/or signaling pathways that participate in axonal pathfinding during embryonic development, and in regeneration, e.g. the CXCL12-CXCR4 axis (Lieberham et al., Neuron 2005), will be included in the discussion. Although this is a very interesting and important axis, it goes beyond the scope of our review, as we are trying to narrow down the molecules to the canonical guidance cues, in order to not to do such an extensive review.
  • When speaking about the difference in regeneration competence of CNS axons vs PNS ones, please include and comment the studies by Hellal et al., Science 2011; Tom et al., J Neurosci. 2004; Ertük et al., J Neurosci. 2007, which deal with the formation of retraction bulbs, and highlight the importance of microtubule stabilization of axons and growth cones (occurring in the PNS and not in the CNS) in order to counteract regeneration failure. The three references were added.
  • Language editing required. We revised and made changes to the English.

Round 2

Reviewer 2 Report

This revised version has been largely amended according to my comments, and there is no concern in the present manuscript except for the following minor comments:

  1. Line 24: not “Spinal cord injury” but “spinal cord injury”.
  2. Line 90: not “cytochalasin B-induces” but “cytochalasin B induces”.
  3. Page 3: please put the legend of Fig. 1 just below Fig. 1.
  4. Line 119: not “enter” but “enters”?
  5. Line 199: is English OK? Not “were” but “where”?
  6. Line 202: not “filopodia” but “fiopodia,”?
  7. Line 246: not “present” but “presents”?
  8. Line 254: not “is not surprising” but “it is not surprising”.
  9. Line 299: not “persist” but “persists”.
  10. Line 322: not “determine” but “determines”.
  11. Line 355: not “binds” but “bind”?
  12. Lines 361-363: not “bind” but “binds”.
  13. Line 420: not “an” but “a”?
  14. Line 430: not “to” but “in”?
  15. Line 709: not “reduce” but “reduces”.
  16. Line 777: not “generates” but “generate”.
  17. Line 797: not “Hashimoto” but “Hashimoto et al.”.
  18. Line 804: not “suggest” but “suggests”?
  19. Line 840: not “show” but “shows”.
  20. Lines 855 and 856: please check English in this sentence.
  21. Line 859: not “affect” but “affects”.
  22. Line 896: not “Lamprey” but “lamprey”.
  23. Line 911: not “binding” but “to bind”?
  24. Line 913: not “Planarian” but “planarian”.
  25. Line 925: not “show” but “shows”.
  26. Line 949: not “have” but “has”.
  27. Line 951: not “increase” but “increased”?
  28. Line 956: not “show” but “shows”.
  29. Line 1630: is this line deleted? If so, there is no definition of DGC in the text. Please give this definition in line 911.
  30. Figure 2 should be placed near page 12 but not page 16.
  31. List of abbreviation: the words in this list should be sorted alphabetically.
  32. There appear to be more English mistakes than pointed out above. Please check your manuscript very carefully by yourself.

Author Response

We thank you for the time inverted in reviewing our manuscript, all the comments have been very usefull.

We incorporated all the suggestions.

We sorted the list of abbreviation alphabetically.

We checked the manuscrips very carefully a couple of times.